# Attention-Linear Trajectory Prediction

**DOI:** 10.3390/s24206636

**Published:** 2024-10-15

**Authors:** Baoyun Wang, Lei He, Linwei Song, Rui Niu, Ming Cheng

**Affiliations:** 1The National Key Laboratory of Automotive Chassis Integration and Bionics, Jilin University, Changchun 130012, China; wangby22@mails.jlu.edu.cn (B.W.); songlw22@mails.jlu.edu.cn (L.S.); niurui22@mails.jlu.edu.cn (R.N.); 2The Computer Science and Technology at the College of Artificial Intelligence, Beijing Normal University, Beijing 100875, China; mingcheng@mail.bnu.edu.cn

**Keywords:** trajectory prediction, linear, self-attention, autonomous driving

## Abstract

Recently, a large number of Transformer-based solutions have emerged for the trajectory prediction task, but there are shortcomings in the effectiveness of Transformers in trajectory prediction. Specifically, while position encoding preserves some of the ordering information, the self-attention mechanism at the core of the Transformer has its alignment invariance that leads to the loss of temporal information, which is crucial for trajectory prediction. For this reason, we design a simple and efficient strategy for temporal information extraction and prediction of trajectory sequences using the self-attention mechanism and linear layers. The experimental results show that the strategy can improve the average accuracy by 15.31%, effectively combining the advantages of the linear layer and the self-attention mechanism, while compensating for the shortcomings of the Transformer. Additionally, we conducted an empirical study to explore the effectiveness of the linear layer and sparse self-attention mechanisms in trajectory prediction.

## 1. Introduction

Autonomous vehicles precisely predict the trajectories of other traffic participants in the next few seconds to enable safe decision-making, whereas it is challenging to precisely predict the trajectories of agents in complex traffic dynamics scenarios. In recent years, learning-based approaches have been widely used for trajectory prediction [1,2,3,4], where a large number of solutions have emerged based on Transformer [5], which has had successful applications in Natural Language Processing (NLP) and computer vision, among others [6,7,8,9], and is considered to be the most successful solution for extracting the serial correlation.

Trajectory prediction is essentially a time series prediction task and Transformer-based time series analysis solutions have produced several well-known models for time series prediction, such as Informer [10], FEDformer [11], and Autoformer [12]. Among them, Informer [10] proposed the ProbSparse self-attention, which reduces the complexity of the Transformer self-attention mechanism. However, in time series modeling, while the Transformer uses positional encoding techniques to retain some ordered information, the inherent alignment-invariant of the self-attention mechanism inevitably leads to the loss of temporal information [13]. The alignment-invariant of the self-attention mechanism refers to its irrelevance to the order of the input sequence. Specifically, during the calculation of attention scores, the dot-product operations between the queries (Q), keys (K), and values (V) only consider the similarity between vectors without taking into account their positions in the sequence. As a result, even if the order of the input sequence changes, the outcome of the dot-product operation remains essentially unchanged. Additionally, the self-attention mechanism applies global weighted averaging to all input elements, disregarding their positions and only performing weighted summation based on the calculated attention scores. This means that the output vectors do not retain the positional information of the input elements, making the entire process fundamentally independent of the input sequence order.

For NLP, the semantics of the overall sentence are preserved to a large extent, even if some of the words are reordered. However, trajectory prediction requires learning temporal changes of successive points, and order plays the most crucial role. Meanwhile, recent studies have emphasized that information about independence and correlation among multiple variables in time series prediction impacts the prediction results [14,15]. Transformer-based forecasting architectures typically embed multiple variables at the same timestamp into indistinguishable channels, causing multivariables to lose their independence. As a result, the performance of Transformers in long-term time series forecasting (LTSF) has been questioned. Researchers have found that the linear model LTSF-Linear [13] using only linear layers outperforms the complex Transformer and its variants in both performance and efficiency in the task of long-term time series forecasting, but have also found that linear layers in combination with other attentional components lead to degraded performance. Based on this study, iTransformer [16] achieved better results in time series prediction by centering on variables, embedding the entire time series of each variable independently into tokens, capturing multivariate correlations by the self-attention mechanism, and applying a feed-forward network to learn nonlinear representations for each variable token.

Trajectory prediction is technically not a simple LTSF task; LTSF is applicable to time series that have relatively significant trends and periodicity, such as weather, traffic flow, energy, economic, and disease forecasts, and their prediction scales are typically measured in hours. In addition, researchers have pointed out that combining linear layers with attention as well as other components leads to performance degradation [13], and the effectiveness of using linear layers for prediction is related to the size of the look-back window (historical series), with the larger the look-back window (96–720), the better the performance of the linear layer. For trajectory prediction, the argoverse [17] dataset, for instance, has a lookback window of only 20.

Through experiments, we found that a simple linear layer is not capable of the trajectory prediction task. Trajectory prediction generally requires predicting precise trajectories for several seconds in the future and faces more complex challenges because the trajectories of agents on the road are affected by interactions between agents and traffic rules, and their potential future trajectory choices are diverse. Although self-attention loses temporal information due to the inherent alignment invariance, the ability of self-attention to extract serial dependencies is still worthy of attention. We believe that while the linear layers cannot be directly integrated with self-attention components, they can operate in parallel with components like self-attention to extract features from trajectory series and then perform feature fusion. This approach can preserve the order of the sequence inputs, thereby compensating for the Transformer’s limitations in extracting temporal information.

High-precision maps are also crucial in trajectory prediction [18]. Map representations help in accurate trajectory prediction, and most of the current trajectory prediction models use map representations [2,17,18,19,20], where VectorNet extracts the feature representations of the agents and lanes, respectively, using two graph convolutions [21], LaneGCN [20] obtains sparsely connected lane graphs from map topological connections and uses polylines as node representations to obtain high resolution.

We explored the effective application of linear layers in the field of trajectory prediction and proposed a flexible trajectory prediction framework by combining sparse self-attention to extract multivariate correlations and dependencies in the series. The effectiveness of the linear prediction scheme and the sparse self-attention mechanism in trajectory prediction is experimentally verified.

In summary, the contributions of our work include:

To the best of our knowledge, our work is the first to introduce the ability to extract temporal information from linear layers to trajectory prediction. It uses linear layers to process sequence features in parallel with self-attention, extracts the temporal information with a simple linear layer, and maintains multivariate independence. This approach solves the issue of temporal information loss in Transformer-based trajectory prediction schemes caused by self-attention and verifies the effectiveness of using linear layers to extract temporal information in trajectory prediction.

The sparse self-attention mechanism is applied to trajectory prediction, in our work, the multivariate correlation information is considered. On the one hand, the sparse self-attention mechanism extracts dependency relationships among the trajectory sequences. On the other hand, it captures the multivariate correlation information, aiming to achieve accurate predictions.

We conducted a series of empirical studies on the designed trajectory prediction scheme, exploring the effects of using linear layers to extract temporal information, the sparse self-attention mechanism to extract trajectory sequence dependencies and variable correlations, and positional encoding in Transformer on the effectiveness of trajectory prediction. Our work will benefit future research in this area.

## 2. Attention-Linear Trajectory Prediction

We propose the Attention-Linear Trajectory Prediction architecture, which uses self-attention to extract dependencies and correlations of variables in the trajectory series, the linear layer to extract the temporal information of the trajectory series, and a prediction component to decode the multivariate channels independently for direct forward prediction of the trajectory series. At the same time, in order to intuitively explore the potential of different forms of self-attention in trajectory prediction and the ability of the linear layer to model temporal information in the trajectory series, we constructed the network using the basic components.

### 2.1. Problem Formulation

We formulate the trajectory prediction problem as predicting the future position coordinates of the target vehicle based on the observed historical motion information of the agents, i.e., the target vehicle and its surrounding traffic participants. Formally, trajectory prediction is based on the past $T_{p}$ time steps $X=\left\{X_{1},X_{2},\ldots ,X_{T_{p}}\right\}$, predicting the future positions $Y=\left\{Y_{1},Y_{2},\ldots ,Y_{T_{f}}\right\}$ for $T_{f}$ time steps. Here, $\mathrm{Xt}=x_{1}^{t},x_{2}^{t},\ldots ,x_{N+1}^{t}$ represents the information of N + 1 agents at time t, where N + 1 denotes the target vehicle and the N surrounding traffic participants. We represent the geometric properties of the agents’ trajectory using relative positions, transforming the trajectory data into vector information. Therefore, in the model, $x_{i}^{t}=P_{i}^{t}-P_{i}^{t-1}$, where P represents the two-dimensional coordinates (x, y) of agent i at a certain time. At this point, X is denoted as ${\left\{P_{i}^{1}-P_{i}^{0},P_{i}^{2}-P_{i}^{1},\ldots ,P_{i}^{T}-P_{i}^{T-1}\right\}}_{i=1}^{N+1}$, and $Y_{t}=P^{t}$, which represents the predicted trajectory coordinates of the target vehicle at time t. Thus, Y represents the $\left\{P^{T_{p}+1},P^{T_{p}+2},\ldots ,P^{T_{p}+T_{f}}\right\}$ of the target vehicle.

### 2.2. Overview of the Structure

As shown in Figure 1, the Attention-Linear trajectory prediction architecture uses the trajectory series attention module to extract the features and dependencies of the trajectory sequences, as well as the interaction information between the variables. The input to the trajectory series attention module is of shape (B, T, V), and the output is (B, T, D), where B represents the number of agents in each batch, T represents the time steps, and V represents the trajectory variables. The model uses the last time step of the historical features, which is the initial time step for prediction, as the input (B, D) to the feature interaction attention module. This module extracts interaction information between the agents and the local traffic environment. The map representation module extracts structured map features, with input (N, V), where N represents the number of lane nodes, and V here represents the map input variables. The linear module utilizes linear layers to extract features from trajectory sequences in a channel-wise manner and aggregate features from the above modules. Finally, it outputs the predicted trajectory sequence of shape (B, K, Tp, 2), where K represents the K multimodal trajectories output, Tp represents the prediction timestep, and 2 represents the x and y coordinates of the trajectory.

### 2.3. Attention-Linear Components

#### 2.3.1. Trajectory Series Attention

First, we extract the trajectory information of the agents from the scene data, use the relative position to represent the geometrical attributes of the agent’s trajectory, and when the length of the agents’ trajectory sequence is less than the historical time step, fill it with the value 0 and mark the filled elements with a mask. The trajectory series attention module uses variables at the same moment as tokens in extracting dependencies between sequences and uses linear layers to transform the variables to a higher dimensional space, and then models the temporal dependencies using the sparse self-attention mechanism [10] with the following formulas:(1)$$X={\mathrm{Linear}}_{V}^{D}\left(\mathrm{input}\right)$$
(2)$$K=XW^{K}$$
(3)$$V=XW^{V}$$
(4)$$Q=XW^{Q}$$
(5)$$\bar{M}\left(q_{i},K\right)=\max_{j}\left(\frac{{q_{i}k_{j}}^{T}}{\sqrt{d}}\right)-\frac{1}{L_{K}}\sum_{j=1}^{L_{K}}\frac{{q_{i}k_{j}}^{T}}{\sqrt{d}},\left\{q_{i}\in Q,k_{j}\in K\right\}$$
(6)$$A\left(Q,K,V\right)=\mathrm{Softmax}\left(\frac{\bar{Q}K^{T}}{\sqrt{d}}\right)V$$
The original input of the trajectory data is transformed from variable V to D dimensions through an embedding linear projection, which serves as the input X for the self-attention layer, after which X is multiplied by the corresponding weight matrix W to generate the queries (Q), keys (K), and values (V). In the sparse self-attention mechanism, the input $\bar{Q}$ in Equation (Equation 6) is a sparse matrix that only contains the top-u queries selected based on a sparsity metric. The sparsity measurement formula is shown in Equation (Equation 5), where $q_{i}$ represents the i-th query, K represents the set of key vectors, $L_{K}$ is the total number of key vectors, *d* represents the dimension of q and k vectors, and the dot product divided by the scaling factor $\sqrt{d}$ prevents the gradient from vanishing or exploding. The sparsity metric distinguishes between major and minor dot products to generate different sparse query key pairs for each attention head, which reduces the time complexity and avoids serious information loss. In addition, residual connectivity [22] is used in the self-attention part to improve the speed and effectiveness of network training, and Layernorm [23] is used to improve the convergence and training stability of the deep network, and the feed-forward network follows the Transformer’s linear connection.

In order to extract correlation information between variables, the trajectory series attention module uses the same components and inverts embedding, where the overall time series of the same variable is used independently as a token, and instead of modeling the temporal information of the series, self-attention is used to extract the interactions between independent variables. Meanwhile, layer normalization is used for the representation of individual variables, and normalizing variables to a Gaussian distribution reduces the discrepancy caused by inconsistent measurement scales.

#### 2.3.2. Map Representation

The map representation provides geometric and semantic information for trajectory prediction. The input lane-level map is first represented as a set of vectorized data, including lane centerlines and connectivity properties between lanes. We denote lane nodes by r. The lane node is represented as the centerline of each lane in the vectorized data of the map, and N represents the total number of lane nodes in the map. There are four kinds of connectivity characteristics of lane nodes, namely, predecessor, successor, left neighbor, and right neighbor, and the corresponding adjacency matrix $M=\left\{M_{p},M_{s},M_{l},M_{r}\right\}$ is constructed from them, where M is an $R^{N\times N}$ matrix, e.g., if node *i* is the right neighbor of *j*, the jth column of the ith row of $M_{r}$ is 1, otherwise it is 0.

Trajectory prediction generally uses the graph convolution operator [20] to process the map information, i.e., graph convolution is used to process the input features of lane nodes and adjacency matrix. Firstly, feature X of the lane nodes is:(7)$$X=\left\{\left.\left(r_{i}^{0}-r_{i}^{1}\right)+r_{i}^{\frac{1}{2}}\right|\;i\in \left(1,N\right)\right\}$$
where $\left(r_{i}^{0}-r_{i}^{1}\right)$ denotes the start-to-finish vector of the lane nodes, which represents the geometry features of the lane nodes, and $r_{i}^{\frac{1}{2}}$ denotes the centroid location features of the lane nodes, and then X is converted to a high-dimensional feature matrix by MLP.

The adjacency matrix represents the connectivity information of the lanes, and the lane node features, together with the adjacency matrix features, constitute the basic elements of the graph convolution operator, and the multi-scale graph convolution can capture the map information in a larger horizon, which is calculated as follows:(8)$$I=\mathrm{XW}+M_{l}XW_{l}+M_{r}XW_{r}+M_{p}^{k}XW_{p}+M_{s}^{k}XW_{s}$$
where W denotes the corresponding weight matrix, which participates in the graph convolution operation with the adjacency matrix M and the lane node features matrix X in training and continuously updates its parameters, *k* represents the multiscale factor, which enables graph convolution of the lane connection features in different perception domains. In addition, residual connections are introduced in the part of graph convolution to improve the speed and effectiveness of network training.

#### 2.3.3. Interactive Attention

Interactive attention is based on spatial distance information, and its main goal is to capture spatial interactions in a traffic scenario. Its scenario embedding inputs are concatenated based on distance queries to road features near the agents as well as other traffic participants. Notably, in contrast to many previous approaches, Attention-Linear avoids using trajectory series features between agents as interactions and instead uses variable features along with sequence features as the target for encoding information about agents’ interactions.

#### 2.3.4. Linear Module

The linear layer itself does not directly capture temporal information but extracts features by applying the same linear transformation to each time step t across different variable sequences. Its advantages lie in weight sharing, simple and effective feature extraction, and the ability to integrate with other related structures. Time series often exhibit certain patterns and regularities, and weight sharing can better capture these patterns, thus playing a crucial role in time series data processing tasks. Additionally, the introduction of the LTSF-Linear [13] model provides valuable insights into our understanding of time series representation. As shown in Figure 2, we designed a linear layer to learn the temporal information representation of the trajectory sequence.

We first perform layer normalization on the entire sequence of variables, followed by independent linear transformations on the entire sequence for each variable, allowing each variable to learn specific feature representations. Within the sequence of the same variable, the linear layer can utilize shared weights across the time dimension. Additionally, since the time steps are processed independently, the feature representations at each time step maintain consistency. Compared to time-step-specific feature transformations like the self-attention mechanism, this approach can extract stable global features from the sequence. The calculation formula is as follows:(9)$${\hat{X}}_{i}={\left\{WX_{i}\right\}}_{i=1}^{V}$$
where W is the linear layer along the time axis, and ${\hat{X}}_{i}$ as well as $X_{i}$ represent the linear transformation feature and input of the ith variable. ${\hat{X}}_{i}$ contains the temporal order of the trajectory sequence. To facilitate efficient learning, common deep learning techniques and residual connections are used. Finally, the features from the interaction attention mechanism are aggregated through simple summation.

In the prediction phase, direct forward prediction is performed, where the sequence dimension is converted to the length of the future prediction sequence through the linear layer. The same linear transformation is applied to each channel. At this point, the weight of the linear layer is shared across all variables, ensuring that the highly correlated output time series of variables x and y are mapped in the same manner. The prediction module consists of K-stacked linear prediction layers. For each traffic participant, K possible future trajectories can be predicted. The linear layer is also directly applied in the prediction section to map the input features to a one-dimensional space as a classification confidence score to predict the prediction closest to the true value among the K trajectories.

### 2.4. Training

**Efficiency analysis:** At this point, we conduct a simple efficiency analysis of the trajectory sequence feature extraction part, trajectory series feature extraction is completed by the trajectory series attention and linear module. The trajectory series attention uses the ProbSparse self-attention mechanism to parallelly extract sequence correlation and variable correlation. The computational complexity related to the sequence length L for the ProbSparse self-attention mechanism [10] is O(Llog L). Therefore, the computational complexity for extracting sequence correlation and variable correlation is O(Llog L) and O(Vlog V), respectively. Since V is smaller than L, the computational complexity of attention can be simplified to O(Llog L). The linear module performs linear transformations on the sequences corresponding to the variables through V linear layers, with a computational complexity of O(L), Therefore, the computational complexity related to the sequence length for the trajectory series feature extraction part can be simplified to O(Llog L).

**Loss calculation**: In the trajectory prediction training phase, we train the whole model end-to-end. For the trajectory prediction task, the loss function *L* consists of two terms: the trajectory regression loss $L_{\mathrm{reg}}$ and the classification loss $L_{\mathrm{cls}}$, defined as shown in Equations (Equation 10)–(Equation 12):(10)$$L=L_{cls}+L_{reg}$$
(11)$$L_{reg}=SmoothL_{1}\;loss\left(t_{j}-t_{gt}\right)$$
(12)$$L_{\mathrm{cls}}=\frac{1}{K-1}\sum_{i\neq j}\max\left(0,p_{i}-p_{j}+m\right)\;i,j\in \left(1,K\right)$$
where the model predicts K trajectories and the trajectory regression loss uses the smoothed $L_{1}$ loss. When calculating $L_{\mathrm{reg}}$, the $L_{2}$ error between the K predicted trajectories and the ground truth trajectory is first calculated. Then, the trajectory with the smallest final displacement error is selected to calculate the loss. i and j both range from 1 to K. $t_{i}$ and $p_{i}$ represent the predicted trajectory and the confidence probability, respectively. m is the maximum margin value. The classification loss begins to accumulate when the probability of the trajectory that is closest to the ground truth is not higher than the probabilities of other trajectories by at least the margin value m; otherwise, the classification loss is zero. j is the index of the trajectory with the lowest average displacement error (ADE) closest to the ground truth trajectory $t_{gt}$:(13)$$j=\arg\min_{i}ADE\left(t_{i},t_{gt}\right)$$

## 3. Experiments

### 3.1. Experimental Settings

**Dataset.** The effectiveness of our method is validated on the Argoverse dataset, a large-scale dataset from real-world sources widely used in the field of trajectory prediction, covering 324,557 diverse traffic scenarios in the Pittsburgh and Miami regions of the U.S. The trajectories of each scenario are recorded via sequential frames sampled at 10 Hz. Each scenario focuses on a specific agent object and includes the historical trajectories of all traffic participants over 2 s and the true future trajectories for the next 3 s. This dataset provides a reliable empirical basis for motion prediction studies in complex real-world environments.**Metrics.** We computed the most widely used evaluation metrics in the field of trajectory prediction:

**min${\mathrm{ADE}}_{k}$**: The average $l_{2}$ distance between the predicted trajectory and the ground truth trajectory across all timesteps.

**min${\mathrm{FDE}}_{k}$**: The $l_{2}$ distance between the endpoint of the predicted trajectory and the ground truth endpoint position.

${\mathrm{MR}}_{k}$: The proportion of predictions where the distance between the predicted endpoint and the ground truth endpoint exceeds the 2-meter threshold.

k represents the number of predicted trajectories. We use k = 6 and k = 1 as performance evaluation metrics. When k = 1, one of the six predicted trajectories is randomly selected to calculate the evaluation metric. When k = 6, the trajectory with the smallest final displacement error is selected to calculate the evaluation metric.

**Training.** We trained the model using 205,942 scenarios, validated it on 39,472 scenarios, and tested it on 78,143 scenarios. The model was trained on an RTX3090 with a learning rate of $1.0\times 10^{-3}$ for 8 epochs, then the learning rate was reduced to $1.0\times 10^{-4}$ and continued up to 36 epochs with a batch size of 32 and using the Adam optimizer.

### 3.2. Experimental Results


**Comparison with Different Methods:** We compared the Attention-Linear model with several baseline methods and some state-of-the-art approaches on the Argoverse [17] test set. The results for K = 1 are reported in Table 1. The baseline methods include the constant velocity baseline [17], nearest neighbor retrieval NN+map [17], Social-LSTM encoder-decoder model LSTM ED+Social [17], goal-driven multimodal trajectory prediction TNT framework [24], a novel vehicle motion prediction method based on multi-head attention Jean [2], an attention-based method using recurrent graphs WIMP [25], a hierarchical graph neural network VectorNet [19], a method combining graph convolution with multi-head self-attention CRAT-Pred [26], and SceneTransformer [27], which uses a masking strategy as model queries. It can be observed that the Attention-Linear model achieves lower minADE and minFDE than the aforementioned methods when K = 1.**Importance of each module:** We analyzed the overall architecture of the Attention-Linear model, studied the function of each module separately, and analyzed its impact on the model’s functionality by reducing specific modules to record the corresponding experimental effects. The corresponding results are recorded in Table 2. Where TSA denotes the part of the Trajectory Series Self Attention module that extracts series features, iTSA denotes the part of the Trajectory Series Self Attention module that extracts correlations between variables, Map represents the Map Representation module, and LinearM denotes the Linear module.


The results in the table indicate that each module of Attention-Linear has improved the model performance, which shows the effectiveness of the overall architecture. Additionally, under the three evaluation metrics for k = 1 and k = 6, the performance improvements of the LinearM model reached 15.31%, 11.10%, and 9.40%, as well as 18.91%, 25.14%, and 39.15%, respectively. The prediction performance of the model can reach the best when combining the advantages of the self-attention mechanism and the linear layer to extract the trajectory series features in parallel. Secondly, the experimental results show that using trajectory series self-attention to extract correlation features between series variables has a certain degree of influence on trajectory prediction.


**Importance of particular components in the module:** In order to further verify the improvement of LinearM on the ability of Transformer to extract temporal information, we add positional encoding to the embedding part of TSA as a comparison. At the same time, we do the effectiveness analysis of the sparse self-attention and change it to the normal self-attention for the comparison experiments. The corresponding results are recorded in Table 3.


From the experimental results, we can see that the effect of sparse self-attention is better than that of normal self-attention, the combination of LinearM and self-attention is better than the combination of Positional encoding and self-attention, and there is a significant improvement in the effect when LinearM is added to the combination of Positional encoding and self-attention, indicating that LinearM can effectively extract the temporal information to improve the performance of the overall model. Although Positional encoding can encode the temporal information of the trajectory series, the temporal information will be lost due to the influence of the self-attention mechanism. LinearM can effectively solve this problem.


**Qualitative results:** As shown in Figure 3, Figure 4, Figure 5 and Figure 6, we present the qualitative results of the Attention-Linear model on the Argoverse validation set. It is evident that the Attention-Linear model can accurately predict agents’ behavior in complex traffic interaction scenarios with a multimodal approach. For clarity, we visualized the centerlines of lanes (gray lines), the true trajectory of one agent (red line), the predicted trajectory (light blue line), and the real positions of other traffic participants at the final moment (gray dots) in each scenario.


## 4. Conclusions

We question the effectiveness of Transformer in trajectory prediction. For the first time, we introduce the capability of linear layers to extract temporal information into the field of trajectory prediction. We use linear layers in parallel with sparse self-attention to process sequential features, then fuse the features in separate channels to ensure the independence of the variables and perform direct forward prediction. Empirical studies show that our strategy of parallel feature extraction using linear layers and a sparse self-attention mechanism can improve the performance of the Transformer in trajectory prediction. Additionally, it validates the effectiveness of sparse self-attention in extracting trajectory sequence dependencies and variable correlations for trajectory prediction.

## Figures and Tables

**Figure 1 sensors-24-06636-f001:**
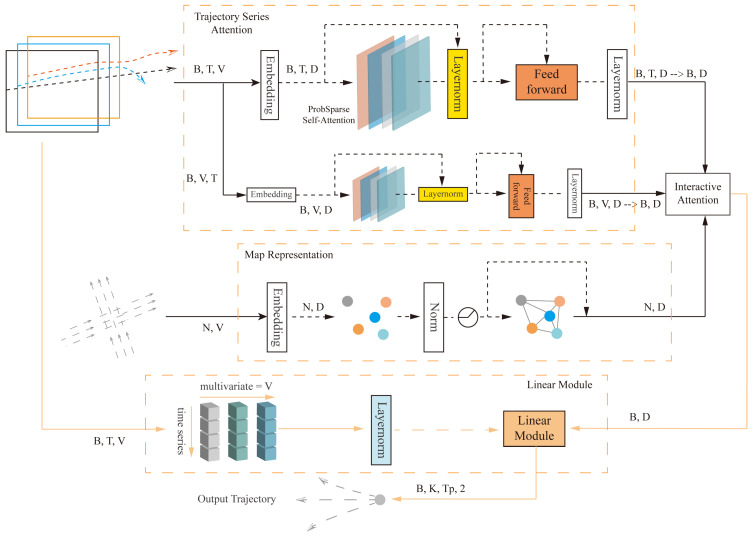
Attention-Linear general network structure.

**Figure 2 sensors-24-06636-f002:**
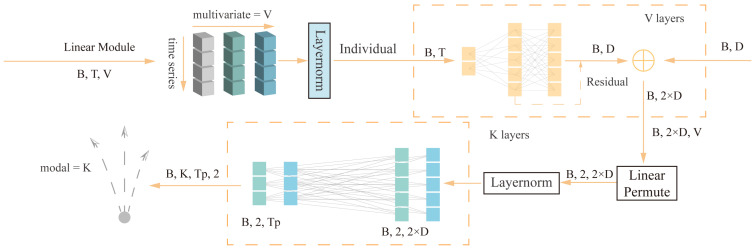
Linear Module Schematic.

**Figure 3 sensors-24-06636-f003:**
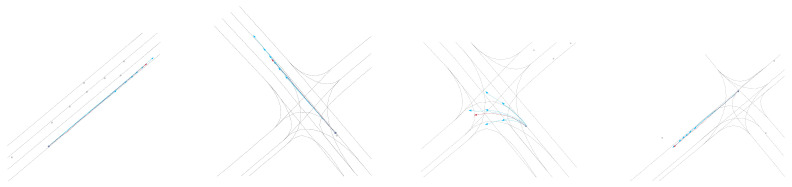
Visualization of the qualitative results for trajectory prediction using the Attention-Linear architecture.

**Figure 4 sensors-24-06636-f004:**
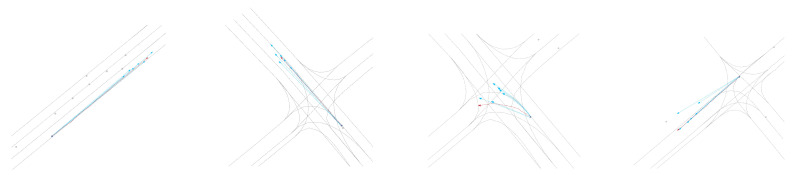
Visualization of the qualitative results for trajectory prediction without LinearM.

**Figure 5 sensors-24-06636-f005:**
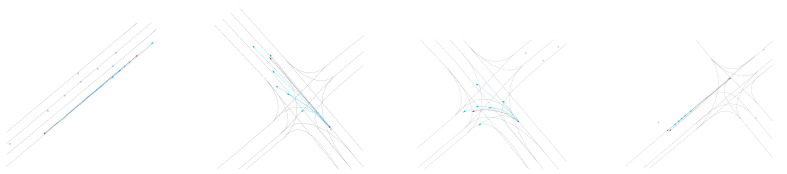
Visualization of the qualitative results for trajectory prediction without iTSA.

**Figure 6 sensors-24-06636-f006:**
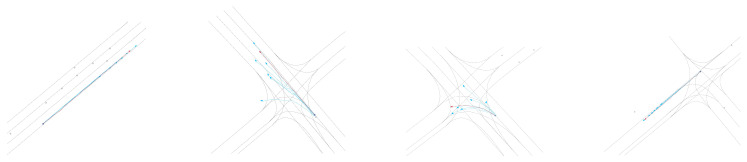
Visualization of the qualitative results for trajectory prediction using Normal-Att.

**Table 1 sensors-24-06636-t001:** Comparison with different methods.

Methods	min${\mathrm{ADE}}_{1}$	min${\mathrm{FDE}}_{1}$	${\mathrm{MR}}_{1}$
Constant Velocity [17]	3.550	7.890	-
NN+map [17]	3.454	7.882	0.871
LSTM ED+social [17]	2.290	5.220	0.680
TNT [24]	2.174	4.959	0.709
Jean [2]	1.860	4.171	0.685
WIMP [25]	1.823	4.030	0.628
VectorNet [19]	1.810	4.010	-
CRAT-Pred [26]	1.816	4.057	0.623
SceneTransformer [27]	1.810	4.055	**0.592**
Attention-Linear	**1.792**	**3.967**	0.611

**Table 2 sensors-24-06636-t002:** Ablation study results 1.

Model Configuration				Final Evaluation Metrics					
**TSA**	**iTSA**	**Map**	**LinearM**	**min${\mathrm{ADE}}_{1}$**	**min${\mathrm{FDE}}_{1}$**	${\mathrm{MR}}_{1}$	**min${\mathrm{ADE}}_{6}$**	**min${\mathrm{FDE}}_{6}$**	${\mathrm{MR}}_{6}$
✓	✗	✗	✗	2.138	4.725	0.694	1.367	2.767	0.451
✗	✗	✗	✓	2.407	5.469	0.712	1.150	2.229	0.340
✓	✓	✓	✗	1.803	3.803	0.617	0.957	1.655	0.212
✓	✗	✓	✓	1.544	3.411	0.561	0.785	1.270	0.136
✓	✓	✓	✓	**1.527**	**3.381**	**0.559**	**0.776**	**1.239**	**0.129**

**Table 3 sensors-24-06636-t003:** Ablation study results 2.

Component Configuration				Final Evaluation Metrics		
**LinearM**	**Positional-En**	**Sparse-Att**	**Normal-Att**	**min${\mathrm{ADE}}_{1}$**	**min${\mathrm{FDE}}_{1}$**	${\mathrm{MR}}_{1}$
✓	✗	✗	✓	1.699	3.710	0.595
✓	✗	✓	✗	1.527	3.381	0.559
✗	✓	✓	✗	1.619	3.471	0.587
✓	✓	✓	✗	**1.511**	**3.357**	**0.557**

## Data Availability

Publicly available datasets were analyzed in this study. This data can be found here: [https://github.com/argoverse/argoverse-api, https://argoverse.org/].

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
