# Peer review of "Attention-Linear Trajectory Prediction"

_sensors, 2024, doi:10.3390/s24206636_

Round 1
Reviewer 1 Report
Comments and Suggestions for Authors
This paper presented a framework of trajectory prediction method by integrating attention mechanism and linear layers. Overall the paper is fairly written and the method is presented clearly. My specific concerns are:
1. The authors used k-stacked linear layers for prediction and selected the best performed prediction results for evaluation. Does this evaluation standard apply to all other baseline models? How can the model be applied to real-time prediction problem? I think the average prediction accuracy should also be provided for all k predictions and compare with baseline models.
2. Did the authors explained the training and test dataset? Is the 205,942 sample used for training and the rest for testing? Please clarify.
3. The formulation needs to be improved, e.g., in section 3.1, it is difficult to understand each variable. Please examine the writing and formulation.
4. Please improve Figure 1 to make it clear module by module.
Comments on the Quality of English Language
The English written is fair, there is rooms for improvement, particularly the conclusion section, which needs highlighting.
Author Response
Thank you very much for your thoughtful and comprehensive reviews of our work.
Please see the attachment.

Reviewer 2 Report
Comments and Suggestions for Authors
This paper a simple and efficient strategy for temporal information extraction and prediction of trajectory sequences using the self-attention mechanism and linear layer. The paper is well-written. There are some comments.
1. The description of the results in the abstract is too subjective. It is suggested that the authors add some quantitative indicator analysis.
2. The existing related work is too simplistic and lacks logical coherence. Suggest the authors to expand and improve this section.
3. This paper proposes a simple algorithm, and to further highlight its main contribution, it is recommended that the author analyze the efficiency of the algorithm.
4. The authors compared many benchmark methods with the proposed method, but the parameter settings of the benchmark methods need to be provided in the paper.
5. This paper only used one database to validate the effectiveness of the proposed model. It is recommended to use other datasets to validate the proposed model.
Comments on the Quality of English Language
Moderate editing of the English language is required.
Author Response

(The authors gave the same response as above.)

Round 2
Reviewer 2 Report
Comments and Suggestions for Authors
No further comments.